# Sample Efficient Reinforcement Learning from Human Feedback via Active Exploration

## Abstract

Preference-based feedback is important for many applications in reinforcement learning where direct evaluation of a reward function is not feasible. A notable recent example arises in reinforcement learning from human feedback (RLHF) on large language models. For many applications of RLHF, the cost of acquiring the human feedback can be substantial. In this work, we take advantage of the fact that one can often choose contexts at which to obtain human feedback in order to most efficiently identify a good policy, and formalize this as an *active contextual dueling bandit* problem. We give an upper-confidence-bound style algorithm for this problem and prove a polynomial worst-case regret bound. We then provide empirical confirmation in a synthetic setting that our approach outperforms existing methods. After, we extend the setting and methodology for practical use in RLHF training of large language models. Here, our method is able to reach better performance with fewer samples of human preferences than multiple baselines on three real-world datasets.

## 1 Introduction

The alignment of foundation models with user preferences has gained unprecedented importance due to the widespread utilization of large language models (LLMs). The established pipeline for alignment in LLMs, as outlined in Stiennon et al. (2020) and Ouyang et al. (2022), comprises two essential steps given a pretrained LLM. First, in the Supervised Fine-Tuning (SFT) phase, the LLM undergoes fine-tuning via supervised learning with examples demonstrating the desired behavior. In the second step, Reinforcement Learning from Human Feedback (RLHF), a policy generates multiple completions for each conversation prefix (prompt) in a training set; users then give ordinal preferences amongst the set of completions for a particular prompt. These preferences are used to train a 'reward model' via a ranking loss like the Bradley-Terry-Luce (BTL) model (Bradley & Terry, 1952). Finally, the policy is trained, typically via Proximal Policy Optimization (PPO) (Schulman et al., 2017), to optimize the reward model while not moving too far from the SFT-trained policy.

As these models continue to scale and their areas of application broaden, the number of roles for which we need to align them increases as does the overall scale of human-generated training data requirements. Data annotation for preference-based learning is already a substantial cost for companies that train LLMs. This cost is likely to grow alongside the industry. This is especially acute for LLMs in specialized areas, where the cost of expert feedback can be substantially higher.

In this work, we take advantage of the fact that we control which prompts and completions we provide to human labelers in order to make efficient use of their efforts. Drawing on recent advancements in active exploration for reinforcement learning (Li et al., 2023) and in black-box optimization (Xu et al., 2020), we introduce a method for assessing the value of collecting preferences on specific datapoints that is both prospective and task-focused. First, we formalize this setting as a *dueling contextual bandit problem* and design an efficient algorithm that offers polynomial worst-case sample complexity guarantees regarding the policy's performance. Next, we extend these ideas to a more real-world setting: choosing datapoints for the training of LLM assistants. Here, we build atop recent work (Rafailov et al., 2023), which allows us to apply active data selection to an RLHF process using a supervised objective and single model. We evaluate the method on three datasets: the Stanford Human Preferences dataset (Ethayarajh et al., 2022), the Anthropic Helpful-Harmless dataset (Bai et al., 2022), and a third dataset (which we contribute to the literature) that extends an existing dataset of Jeopardy! questions and answers to evaluate the ability of an alignment method

to avoid hallucinations. We find that our algorithm can boost performance by over 10% on the preference datasets when performing RLHF with a modest human-feedback sample budget, and that our method is best at avoiding hallucinations on our Jeopardy! dataset.

## 2 RELATED WORK

**Learning from Comparative Feedback**   Reinforcement learning from comparative human feedback has been studied for more than a decade, including work by Fürnkranz et al. (2012), Akour (2014) and, notably, Christiano et al. (2017), which enabled sample-efficient collection of human feedback for deep reinforcement learning (RL) by training a reward model as the RL target. In the Atari test case, where naive deep RL would have necessitated thousands of hours of gameplay, they accomplished superior performance with just 5,500 or several hours of human queries.

Many recent approaches have demonstrated the effectiveness of using human feedback to enhance stylistic continuation (Ziegler et al., 2019), text summarization (Stiennon et al., 2020), translation (Kreutzer et al., 2018), semantic parsing (Lawrence & Riezler, 2018), review generation (Cho et al., 2018), and evidence extraction (Perez et al., 2019). In particular, the work by Bai et al. (2022) places focus on improving model reliability and robustness by incorporating human feedback to gauge the helpfulness or harmfulness of its responses. However, while effective, the integration of human feedback comes with substantial costs. For example, Stiennon et al. (2020) achieved substantial enhancements over baseline methods but required the generation of summaries for 123,169 posts from the TL;DR dataset, a task performed by a large team of labelers from crowdsourcing platforms. This heavy-resource requirement is reflected in state-of-the-art work. Ouyang et al. (2022) emphasizes RLHF to improve alignment of the GPT-3 model across aspects such as toxicity, hallucinations, moral opinion, and overall quality. Here, the team enlisted the efforts of 40 labelers and worked with a dataset comprising over 100,000 examples labeled by humans.

**Dueling Bandits**   The bandit literature has also explored the effectiveness of comparative feedback—for example, in the "dueling bandit" setting—while considering the cost of acquiring such information. This was first studied by Yue et al. (2012) in settings where comparative information is relatively easy to extract but absolute rewards (*i.e.*, direct queries) are ill-defined and have no absolute scale. Later, Bengs et al. (2021) surveyed methods used in the online learning setting, where the trade off with cost of information is most acute, including those used in the online contextual dueling bandit setting by Dudík et al. (2015). These constraints motivate a kernelized approach that can incorporate the nonlinearities in the models used in practice.

**Active Contextual Bandit Optimization**   When there exist distinct phases of learning and then deployment, an agent can often take steps for improved sample efficiency. For example, in a contextual bandit setting, Char et al. (2019) consider the problem where at test time the goal is to perform well on average across a context distribution, while during the learning phase the goal is to choose both contexts and actions for best performance at test-time. The authors proposed a multi-task version of Thompson sampling during the training phase, which yields provable regret bounds. We extend this setting from cardinal to ordinal rewards as is appropriate for comparative feedback.

In Li et al. (2023), the agent queries contexts where the value function is most uncertain and acts optimistically. Combined with least-squares value iteration, this method leads to provable polynomial-sample convergence in the worst-case error of the value function estimate in reinforcement learning in general, and as a corollary the setting from Char et al. (2019) as a special case. This sets the foundation that we will adapt to the comparative feedback setting.

In the realm of flactive contextual bandits that make use of kernels, previous research has explored various aspects, including robust objectives (Bogunovic et al., 2018), distributional robustness (Kirschner et al., 2020; Ramesh et al., 2023), multi-agent learning and mixed strategies (Sessa et al., 2019; 2020). However, to our knowledge, none of the methods proposed in these prior studies can be directly employed in our specific dueling setting.

We also include related work on uncertainty estimation in large language models in Sec. F.

## 3 PROBLEM SETTING

In this paper, we consider a dueling variant of what we denote the active contextual bandit problem introduced in Char et al. (2019) that we refer to as ACDB for short. An instance of this problem

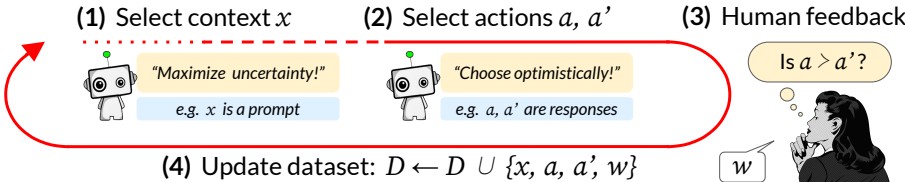

Figure 1: Illustration of the active contextual dueling bandit setting, and its application to sample-efficient RLHF in large language models.

is defined by a tuple $(\mathcal{X}, \mathcal{A}, f)$ where $\mathcal{X}$ denotes the context space, $\mathcal{A}$ denotes the action space and $f : \mathcal{X} \times \mathcal{A} \times \mathcal{A} \to [0,1]$ is a preference function so that $f(x, a, a')$ denotes the probability that the action $a$ is preferred to the action $a'$ when the underlying context is $x$. We also define a domain $\mathcal{D} = \mathcal{X} \times \mathcal{A}$. We will design algorithms that operate under the following interaction protocol, which occurs for $T$ time steps. During each time step $t \in [T]$, the agent selects a context $x_t \in \mathcal{X}$ and a pair of actions $a_t, a'_t \in \mathcal{A}$ and observes a binary random variable $w_t \sim \mathrm{Bern}(f(x_t, a_t, a'_t))$ which equals one if $a_t$ is preferred to $a'_t$ and zero otherwise.

We assume that the preference function takes the following form,

$$f(x, a, a') = \rho\left(r(x, a) - r(x, a')\right), \tag{1}$$

where $\rho : \mathbb{R} \to [0, 1]$ is the *link function* and $r : \mathcal{D} \to \mathbb{R}$ is the *unknown* reward function. Common link functions include the logistic function, which leads to the Bradley-Terry-Luce (BTL) model (Bradley & Terry, 1952) as well as the Gaussian CDF (Thurstone, 1927). We also place some additional assumptions on the reward function for our theoretical analysis in the kernelized setting.

Our objective within this protocol is to design algorithms that are able to quickly identify policies with a small suboptimality gap. We define the suboptimality gap of a learner's policy $\pi : \mathcal{X} \to \mathcal{A}$ as

$$\mathrm{SubOpt}(\pi) = \sup_{x \in \mathcal{X}} \left( \sup_{a \in \mathcal{A}} r(x, a) - r(x, \pi(x)) \right). \tag{2}$$

We remark that this notion of suboptimality (considered in Char et al. (2019) and Li et al. (2023)) is stronger than usual notions that look at the expected suboptimality of the final policy when the contexts are sampled from some known distribution. In contrast, the form of suboptimality we consider here looks at the worst-case context for each policy. For the kernelized and LLM settings we address below, we will make explicit the instantiation of this problem setting.

## 4 ACTIVE EXPLORATION IN THE KERNELIZED SETTING

In this section, we describe our first contribution—a theoretically principled algorithm for the ACDB problem—and provide formal guarantees on its performance. In order to provide such guarantees, we must first instantiate our general problem setup by making assumptions on the preference function $f$ (from Eq. (1)). In particular, we need to specify a class of functions that contain the true unknown reward function. This choice is subtle as we need to balance the trade-off between the expressiveness of our function class with theoretical tractability. Motivated by its theoretical popularity and empirical success, we choose this function class to be a Reproducing Kernel Hilbert Space. While this choice of function class is common in the literature, we take a slight departure from the standard assumptions in order to more appropriately accommodate our problem setting.

**The Contextual Borda Function**   Before going over our assumptions, we first introduce the *contextual Borda function* $f_r$, which is core to our algorithm. The contextual Borda function generalizes the Borda function introduced in Xu et al. (2020) for dueling-choice optimization which is defined as the probability that a particular action $a$ will be preferred over a random action $a'$ uniformly sampled from the action space. We generalize this definition to the contextual setting as follows, given as $f_r : \mathcal{D} \to [0,1]$ where $f_r(x, a) = \mathbb{E}_{a' \sim U(\mathcal{A})}[f(x, a, a')]$, where $U(\mathcal{A})$ is the uniform measure over the action space. It is clear from the definition that $f_r$ and $r$ will have the same maximizers.

We now discuss the assumptions we make. Our first assumption restricts the reward and contextual Borda functions to be 'smooth' in an underlying Reproducing Kernel Hilbert Space (RKHS).

**Assumption 1.** *Let $\kappa : \mathcal{D} \times \mathcal{D} \to \mathbb{R}$ denote a positive semi-definite kernel and let $\mathcal{H}_\kappa$ denote its associated RKHS. We assume that $\|r\|_\kappa, \|f_r\|_\kappa \leq B$, where $B$ is a known constant.*

Note that this assumption is stronger than the standard assumption, which only requires that $r$ has a bounded RKHS norm. It is difficult to bound the norm of $f_r$ given a bound on the norm of $r$ due to the generality of our setting, which allows for different link functions. We investigate this issue numerically in Appendix C where we find that the norm of the Borda function is almost always smaller than the norm of the reward function for samples drawn from the distribution of basis functions used for experiments in Section 4.3.

Our second assumption relates the reward function to the contextual Borda function.

**Assumption 2.** *Let $f_r^*(x) = \max_a f_r(x, a)$ and $r^*(x) = \max_a r(x, a)$. There exists a constant $L_1$ such that for every $x \in \mathcal{X}$, $a \in \mathcal{A}$ we have $\frac{1}{L_1}(r^*(x) - r(x, a)) \leq f_r^*(x) - f_r(x, a)$.*

This assumption implies that differences in $r$ will cause a similar magnitude of difference in $f_r$. In fact, when the link function $\rho(\cdot)$ is Lipschitz continuous, it is sufficient for its Lipschitz constant to be at least $1/L_1$ for this condition to hold. We note that this assumption holds for the two most commonly used link functions, the logistic function (Bradley & Terry, 1952) and the Gaussian CDF (Thurstone, 1927).

### 4.1 METHODS

At a high level, our approach reduces the dueling feedback problem to contextual optimization over a single action via the *contextual Borda function* introduced above. Once reduced appropriately, we apply techniques adapted from recent work on active exploration in reinforcement learning to construct a sampling rule and policy selection rule which allows us to output a policy with provably low sub-optimality. Broadly, our sampling rule samples contexts at which there is maximum uncertainty over the Borda 'value function' and then compares the optimistic action with an action sampled uniformly from the action set.

**Estimating the Contextual Borda Function** By design, we can estimate the contextual Borda function using preference data $\{x_t, a_t, a_t', w_t\}$ by selecting $x_t, a_t$ in an arbitrary fashion and sampling $a_t'$ uniformly at random. For low dimensional settings, our algorithm first estimates the contextual Borda function using standard kernelized ridge regression (KRR) (Rasmussen et al., 2006)—we refer the reader to Appendix A for an explicit description of this regression procedure. In Section 5, we explore modifications of our methods for higher-dimensional settings, such as in the case of LLMs. One key feature of KRR is that it provides both an estimate of the contextual Borda function after $t$ observations, $\mu_t(x, a)$, as well as uncertainty quantification of the predictions. Indeed, under Assumptions 1 and 2 we can show that $|f_r(x, a) - \mu_t(x, a)| \leq \beta\sigma_t(x, a)$ for an appropriately chosen $\beta$ and $\sigma_t(x, a)$ (see Lemma 2).

**Selecting Contexts and Actions** Our sampling rule builds on top of the one established in Li et al. (2023)—put simply, the rule is to sample the state with the maximum uncertainty over the value function and then act optimistically. We now present our algorithm which shows how to extend these ideas to the dueling setting via the contextual Borda function $f_r$.

For now, we assume that there is a known bonus term $\beta_t^{(r)}$ for all $t$. We can then define upper and lower confidence bounds $\overline{f_r^t}(x, a) = \mu_t(x, a) + \beta_t^{(r)}\sigma_t(x, a)$ and $\underline{f_r^t}(x, a) = \mu_t(x, a) - \beta_t^{(r)}\sigma_t(x, a)$. Our rule is to select a context

$$x_t \in \arg\max_{x \in \mathcal{X}} \left( \max_{a \in \mathcal{A}} \overline{f_r^t}(x, a) - \max_{a \in \mathcal{A}} \underline{f_r^t}(x, a) \right). \tag{3}$$

Here, we are choosing a context that maximizes the difference between the optimistic 'value function' and the pessimistic 'value function' (both of which require a maximization over actions to compute). We then optimistically choose an action

$$a_t \in \arg\max_{a \in \mathcal{A}} \overline{f_r^t}(x_t, a). \tag{4}$$

After repeating this process $T$ times, we output a pessimistic policy against the tightest lower bound we can find, which is the maximizer of all our lower bounds through the optimization process. Put formally, we return $\hat{\pi}_T : \mathcal{X} \to \mathcal{A}$ such that

$$\hat{\pi}_T(x) \in \arg\max_{a \in \mathcal{A}} \max_{t \leq T} \underline{f_r^t}(x, a). \tag{5}$$

---

**Algorithm 1** AE-Borda

---

1: **Input:** kernel function $\kappa(\cdot, \cdot)$, exploration parameters $\beta_t^{(r)}$, number of inital data $n_0$
2: Let $D_{n_0} = \{x_i, a_i, a_i', w_i\}_{i=1}^{n_0}$ for $x_i, a_i, a_i'$ drawn uniformly at random.
3: **for** $t = n_0 + 1, \ldots, T$ **do**
4:     Compute $\mu_t(\cdot, \cdot)$, $\sigma_t(\cdot, \cdot)$ using KRR.
5:     Choose $x_t$ according to Eq. (3).
6:     Choose $a_t$ according to Eq. (4), draw $a_t' \sim U(\mathcal{A})$, and draw $w_t \sim \text{Bern}(f(x_t, a_t, a_t'))$.
7:     Let $D_t = D_{t-1} \cup \{(x_t, a_t, a_t', w_t)\}$.
8: **end for**
9: Output a final policy $\hat{\pi}_T$ according to Eq. (5).

---

From these pieces we construct the full active exploration algorithm, AE-Borda, which we present in Algorithm 1.

## 4.2 ANALYSIS

Before proceeding with our algorithm's formal guarantees, we first introduce the *maximal-information gain* which plays an important role in our results. The maximum information gain over $t$ rounds, denoted $\Phi_t$, is defined as

$$\Phi_t = \max_{A \subset \mathcal{X} \times \mathcal{A}: |A| = t} I(r_A + \epsilon_A; r_A), \tag{6}$$

where $r_A = [r(x)]_{x \in A}$, $\epsilon_A \sim N(0, \eta^2 I)$, and $I(X; Y) = H(X) - H(X|Y)$ is the mutual information. With this definition, we are now ready to state our result.

**Theorem 1.** *Suppose we run Algorithm 1 with*

$$\beta_t^{(r)} = 2B + \sqrt{2\Phi_t + 1 + \log\left(\frac{2}{\delta}\right)}, \tag{7}$$

*then, with probability at least $1 - \delta$, we have that*

$$\text{SubOpt}(\hat{\pi}_T) \leq O\left(\frac{L_1}{\sqrt{T}}\left(B + \Phi_T\sqrt{\log\frac{1}{\delta}}\right)\right). \tag{8}$$

**Proof Sketch.** At a high-level, the proof of this result is as follows. First, we use standard results on KRR to show that our choice of $\beta^{(r)}$ guarantees that our confidence bands contain $f^r(x, a)$ with high probability simultaneously for all $t$ and $x, a \in \mathcal{X} \times \mathcal{A}$. Next, we use Assumption 2 to show that the suboptimality of the pessimistic policy induced by our estimated contextual Borda function is small whenever we are able to estimate the contextual Borda function well. Finally, we conclude the proof by showing that our sampling rule indeed allows us to estimate the contextual Borda function well. The full proof can be found in Appendix 1.

**Concrete Performance Bounds.** To more concretely understand the performance of our algorithm, we instantiate our results for two commonly studied kernels: the linear and squared-exponential. For both of these settings, the scaling of the information gain is well known (see for example Srinivas et al. (2010)). In the linear setting, we have that $\Phi_T = O(d \log T)$ leading to a bound of $O\left(\frac{L_1}{\sqrt{T}}(d \log T)\right)$. For squared exponential kernels we have $\Phi_T = O\left(\log(T)^{d+1}\right)$ leading to a suboptimality bound of $O\left(\frac{L_1}{\sqrt{T}}\left(\log(T)^{d+1}\right)\right)$.

When compared to existing results for dueling bandits (Xu et al., 2020) as well as standard bandits (Chowdhury & Gopalan, 2017), we see that our suboptimality bounds match, thus showing that our algorithm is able to achieve the same performance under a stronger performance metric.

## 4.3 EXPERIMENTS IN THE KERNELIZED SETTING

In order to assess the validity of our theory we have conducted synthetic experiments that allow us to come as close as possible to the theoretical setting and empirically confirm our results.

To do so, we implemented the regression using the BernoulliGP model provided by GPyTorch (Gardner et al., 2018). We use a Matérn kernel with automatic relevance detection with hyper-parameters fit via maximum a posteriori optimized by the Adam algorithm (Kingma & Ba, 2014).

We tested on distributions of synthetic reward functions generated by sampling a random linear combination of Random Fourier Features (Rahimi & Recht, 2007) derived from a squared exponential kernel. For each sampled reward function $r$, we used the Bradley-Terry model where $p(w = 1 \mid a, a', x) = \frac{1}{1+\exp(r(x,a')-r(x,a))}$ to generate comparison data. For each trial we uniformly sampled $n_0 = 25$ datapoints and then selected data to observe until $T = 500$ total datapoints had been collected according to one of three methods:

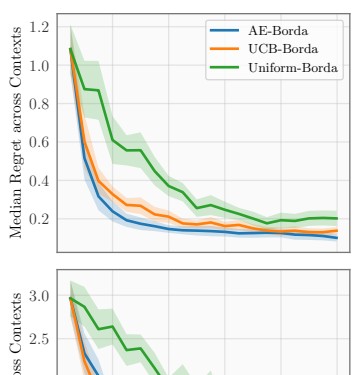
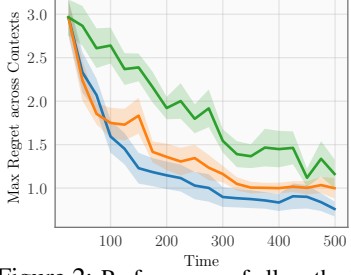

- **AE-Borda**: our method, as described in Section 4.1.

- **Uniform-Borda**: uniform sampling of both contexts and actions.

- **UCB-Borda**: uniform sampling of contexts, along with UCB actions as in AE-Borda.

This last method reduces to the method presented in Xu et al. (2020) when naively generalized to the contextual setting. All methods have the same test-time behavior of executing the action found by optimizing the pessimistic Borda function estimate for the test context. By optimizing the ground-truth reward function we were able to approximate the optimal policy and therefore estimate the regret of our policy against it. We give an example of the progression of our method for 1D context and 1D actions in Figure 3 as well as a comparison against Uniform-Borda and UCB-Borda in Figure 2. One can see that AE-Borda performs best both on median regret and on the maximum regret, which was the metric of interest in our theoretical analysis.

Figure 2: Performance of all methods across 10 random functions $r$ with 1D Context and 1D action. The top plot shows the median regret across contexts and the bottom shows the maximum. Error bands show one standard error.

It is clear in Figure 3 that the method is quickly able to concentrate samples in regions that could plausibly be the optimum and it is similarly clear that the peaks in the acquisition function over contexts are sensible given the mean and uncertainty estimates of $f_r$. We give a set of results showing the progression of AE-Borda in Section D.

## 5 SCALING ACTIVE EXPLORATION TO LARGE LANGUAGE MODELS

In order to adapt our method to the case where $\mathcal{X}$ and $\mathcal{A}$ are both large spaces of sequences as is common in natural language processing, we must address a few limitations of the AE-Borda method presented in Section 4.1:

- The contextual Borda function $f_r$ as defined above is unsuitable for an action space that is extremely large and where most actions are obviously bad (a uniformly sampled sequence of tokens is trivially distinguishable from natural language).

- Neural network training proceeds in batches and it would be highly inefficient to label and train on a single example at a time.

- The uncertainty estimation tools in sequence modeling are more limited than those for explicitly kernelized models, especially due to the memory constraints in training LLMs.

We address these issues through a handful of modifications to our method as we specialize it to the LLM case. Though these modifications mean that we lose the theoretical guarantees in the previous section, we assert that given the rates of convergence associated with kernelized approximations of neural net architectures, we are not giving up strong guarantees in this setting. In particular, we modify the selection rule given in Eq. (3) to avoid having to use the Borda function, we naïvely do batched subset selection for our training minibatches, and we estimate the uncertainty of our policy using dropout for uncertainty estimation (Gal & Ghahramani, 2016). In this section, we build atop the foundation presented in Rafailov et al. (2023), which avoids training a separate reward model;

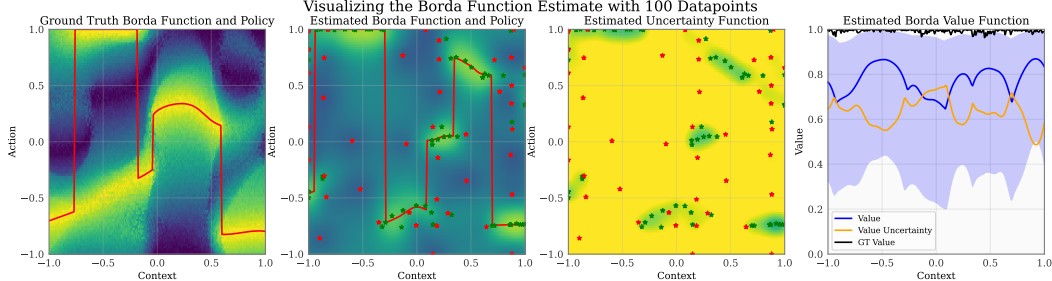

Figure 3: From left: the ground truth contextual Borda function $f_r$ (the red line is the optimal policy), the mean of our posterior estimate of $f_r$ (the red line is the best policy estimate), the uncertainty function $\sigma_t$, and the value function $\max_a f_r^t$. In the middle two plots, red dots are queries where $w_t = 0$ and green are where $w_t = 1$. We plot the value function with confidence intervals in blue on right as well as the value function uncertainty from Eq. (3) in orange. For a full version of this Figure, see Fig. D.

this is primarily due to the fact that we prefer to select datapoints based on the estimated uncertainty of the model used for decision making rather than any proxy.

**Direct Preference Optimization**  Direct Preference Optimization (DPO) (Rafailov et al., 2023) avoids training a separate reward model based on preferences by instead training the policy directly on pairwise comparison using a loss that optimizes an equivalent objective despite functionally behaving like classification. As with PPO (Schulman et al., 2017), this loss depends on a reference policy, which we take to be the policy derived from the supervised fine-tuning step, $\pi_{\text{SFT}}$. The loss is defined as $\mathcal{L}_{\text{DPO}}(\pi_\theta; \pi_{\text{SFT}}) = -\mathbb{E}_{(x,a,a',w)\sim\mathcal{D}} \left[ \log \rho \left( \gamma(2w-1) \left( \log \frac{\pi_\theta(a|x)}{\pi_{\text{SFT}}(a|x)} - \log \frac{\pi_\theta(a'|x)}{\pi_{\text{SFT}}(a'|x)} \right) \right) \right]$. The derivation in Rafailov et al. (2023) also shows that optimizing this objective is equivalent to training a PPO policy with reward function

$$r(x,a) = \gamma \log \frac{\pi_r(a \mid x)}{\pi_{\text{SFT}}(a \mid x)} + \gamma \log Z(x), \tag{9}$$

where $\gamma$ is the hyperparameter of PPO scaling the KL penalty, $Z(x)$ is a partition function, and $\pi_r$ is the policy which optimizes the PPO objective.

**An Acquisition Function for DPO**  We observe as in the original paper that $\pi_r$ is precisely the probability distribution which DPO is estimating. Therefore, the uncertainty estimates for our DPO policy are uncertainty estimates for $\pi_r$ and we can use them to give an approximate confidence interval for $r$ ($\overline{r}$ and $\underline{r}$). Concretely, we need to address the autoregressive nature of $x$ and $a$ in our case. We will assume $a$ consists of ordered tokens $t_i$ and that $\log \pi(a \mid x) = \sum_{t_i \in a} \log \pi(t_i \mid x, t_1, \ldots, t_{i-1})$. In our method, we employ dropout for uncertainty quantification. Specifically, the $m$ dropout masks $d_j$ are integrated into the function $\pi(t_i \mid x, t_1, \ldots, t_{i-1}, d_j)$. During inference, we perform autoregressive Monte Carlo sampling with dropout enabled, resulting in an ensemble of predictions with a mean $\mu(t_i \mid x, t_1, \ldots, t_{i-1}) = \frac{1}{m} \sum_{j\in[m]} \log \pi(t_i \mid x, t_1, \ldots, t_{i-1}, d_j)$. The standard deviation $\sigma(t_i \mid x, t_1, \ldots, t_{i-1}) = \sqrt{\frac{1}{m-1} \sum_{j\in[m]} \left( \log \pi(t_i \mid x, t_1, \ldots, t_{i-1}, d_j) \right)^2}$ across this ensemble serves as an approximation for the model's epistemic uncertainty. This technique allows us to capture uncertainty in a computation and memory efficient manner without compromising model performance. Given these estimates, we can compute our upper and lower bounds as follows:

$$\overline{r}(x,a) = \sum_{t_i \in a} \mu(t_i \mid x, t_1, \ldots, t_{i-1}) + \beta\sigma(t_i \mid x, t_1, \ldots, t_{i-1}) - \log \pi_{\text{SFT}}(a \mid x), \tag{10}$$

$$\underline{r}(x,a) = \sum_{t_i \in a} \mu(t_i \mid x, t_1, \ldots, t_{i-1}) - \beta\sigma(t_i \mid x, t_1, \ldots, t_{i-1}) - \log \pi_{\text{SFT}}(a \mid x), \tag{11}$$

for an uncertainty parameter $\beta > 0$. In the previous section, we chose contexts according to Eq. (3). Here, we define an acquisition function using a similar quantity:

$$\alpha(x) = \max_{a\in\mathcal{A}(x)} \overline{r}(x,a) - \max_{a\in\mathcal{A}(x)} \underline{r}(x,a). \tag{12}$$

---

**Algorithm 2** AE-DPO

1: **Input:** Reference policy $\pi_{\text{SFT}}$, exploration parameter $\beta$, policy constraint weight $\gamma$, batch size $b$, number of iterations $N$
2: **for** $t = n_0 + 1, \dots, N$ **do**
3:     Draw an unlabeled batch $B_u = \{(x_i, a_i, a_i')\} \sim D$.
4:     Evaluate $\alpha(x_i)$ and let $B_l$ be a batch of the top-$b$ elements of $B_u$ by $\alpha$ value.
5:     Collect labels $r_i$ and add them to $B_l$.
6:     Update the policy $\pi_\theta$ (initialized from the ref.) using a gradient step against $\mathcal{L}_{\text{DPO}}$ using $B_l$.
7: **end for**
8: Output $\pi_\theta$

---

In this equation, $\alpha(x)$ is the uncertainty of the state-value function according to $x$. In choosing the states where the potential for error in the value achieved is largest, the agent can learn to behave well in those places. This criterion is similar to that in Li et al. (2023) and provides similar guarantees to ours for max-regret in the active contextual bandit setting. In situations like ours where we are using fixed offline datasets, we set $\mathcal{A}(x)$ in Eq. (12) to the set of available responses for a particular action; otherwise, we use $\mathcal{A}(x) = \mathcal{A}$.

**An algorithm for active RLHF**   From here, we use the acquisition function in Eq. (12) in order to choose points that are maximally informative. We must do this in batches in order to respect the constraints of training large models. We address this in the naive fashion, pulling a larger batch of some size, evaluating $\alpha$ and then choosing the top-$b$ elements in order to address this criterion. We refer to our full procedure as AE-DPO, and give a description in Algorithm 2.

## 5.1   Experiments using LLMs

In order to evaluate whether our method is able to improve the selection of datapoints in RLHF, we conduct a set of experiments in which we train LLMs on three datasets using one of four methods. The goal of our empirical study is to see whether improving the data selection strategy causes the downstream policy to perform better on a given training task. In order to isolate the effect of the data selection method, we vary the selection method while largely holding our model and training procedure consistent. In all the experiments in this section, we compare four methods: **DPOAE**, the method we presented in Section 5; **USDPO**, which chooses $x$ that maximize variance of the log probabilities of completions; **DPO**, the method from Rafailov et al. (2023), selecting batches uniformly at random; and **SFT**, which continues supervised learning with uniformly selected batches. In our training pipeline, we first train a baseline model with a Llama-7B (Touvron et al., 2023) architecture using supervised fine-tuning (SFT) on a 40% split of data. We add a dropout layer before the penultimate linear layer for our uncertainty estimation mechanism and fine tune with dropout active. Next, we train using each of the four methods for 30000 samples, evaluating every 2048 samples— each time using our initial SFT trained model as a starting point. We give additional information on our experimental procedures in Section H.

We evaluate these methods on three different datasets. The first two, the Anthropic Helpful-Harmless (HH) dataset (Bai et al., 2022) and the Stanford Human Preferences (SHP) dataset (Ethayarajh et al., 2022), are taken from the literature. HH contains examples of two kinds: situations where an assistant needs to be helpful to a user asking a reasonable question and situations where an assistant should prioritize being harmless as the user is requesting a harmful action. All completions in HH are machine-generated. SHP is a dataset of Reddit posts with comments in 18 different categories and therefore consists of a broader range of human-generated text, but doesn't have the inherent tradeoff of HH. We evaluate policies trained on both of these by checking the rate at which the policy produces answers which are preferred to the chosen completion for the prompt in the dataset.

For the completions generated from the HH and SHP prompts, we use GPT-3.5 (Brown et al., 2020) to generate winners amongst comparisons between the preferred choices given in the dataset. We give the prompts we use for evaluation in Section G. In Figure 2, we see that for the completions in the later part of our training run, AE-DPO performs best among the methods and outperforms US-DPO as well as the other baselines that sample uniformly. We believe this to be due to our acquisition function $\alpha$, which accounts for the structure of the decision making problem in choosing which point to query. We do find our results to be noisy—due to the computational expense of these trials (which we elaborate on in Section I), we were not able to run each experimental baseline for a large number of seeds to further reduce uncertainty in our results.

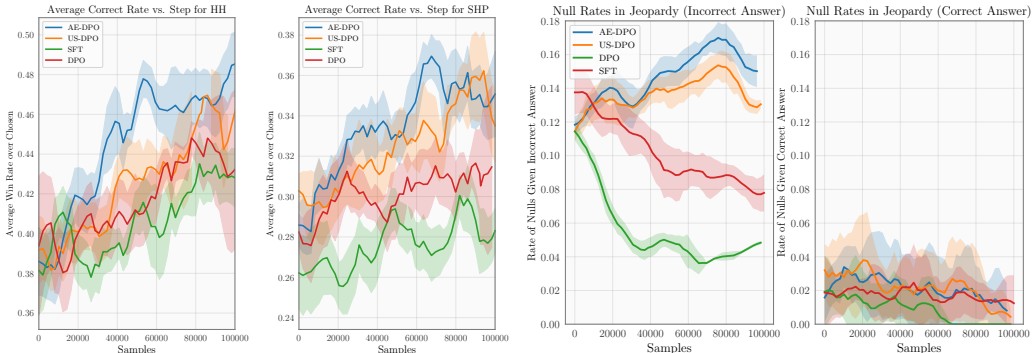

Figure 4: From left: smoothed win rates against preferred choices in dataset of samples generated from each policy at end of RLHF training runs across the final four evaluations, and all seeds, on the HH (first) and SHP (second) datasets. In the latter two plots, we force each policy to generate a (non-null) answer, and then, conditional on the answer being correct (fourth) or incorrect (third), plot the rate at which each policy abstains.

We also introduce a new Jeopardy! dataset that includes a preference structure that captures some of the structure of the game while addressing the question of LLM hallucination. We augment a dataset consisting of 217,000 Jeopardy! questions and answers from HuggingFace (Wolf et al., 2023) with a plausible incorrect answer, using GPT-3.5. As in the game show, where points are deducted for an incorrect response, we enforce during training that a correct answer is preferred to an abstention (the empty string) and both of these should be preferred to the incorrect answer. We found that our models do not learn to provide correct answers at a higher rate through a small amount of DPO training or additional SFT beyond what is required for them to answer the questions. This is unsurprising as trivia is intended not to generalize easily; in other words, it's difficult to imagine learning that the third US president was Jefferson given training examples of the first two. Instead, we evaluate policies for this dataset on the rate at which they abstain for questions ("null rate") where they counterfactually would have been correct vs where they would have been incorrect. Ideally, the policy learned would *always* abstain where it would have been incorrect and *never* abstain where it would have been correct. Naturally, this is an important goal in the alignment of LLMs and we hope to provide a straightforward benchmark for this effort. We include an additional exhibit where we use the factual nature of this dataset to begin to evaluate the dropout-based uncertainty estimation techniques we use in appendix J.

For the Jeopardy! dataset, we checked the probability of an empty generation and whether it was the most likely token. We generated a nonempty sequence in order to see whether the generated answer was correct, including as a counterfactual in the cases where the method would have abstained. We plot this in Figure 4, where we see that the AE-DPO method is the only method that learns to abstain from answering questions (modestly) more often when the model would have given the incorrect answer. We also find that the standard DPO method quickly learns not to abstain. No methods abstain more than a couple percent of the time in the case where they would have been correct. We also plot the results for correctness in Section J, which shows that no model substantially learns new factual information.

## 6 DISCUSSION

In this work, we addressed the problem of how to select contexts and actions at which to obtain human preferences, such that the reinforcement learning agent learns most efficiently. We focus on this problem setting in the context of reinforcement learning from human feedback in large language models (LLMs), where collecting data from humans is expensive. This problem is particularly meaningful because, in the future, it is likely that we need feedback from specialized humans whose time is extremely limited and in order to provide personalization of LLMs without a prohibitive training period. The methods developed in this work show promise in reducing these costs. We also make a theoretical contribution where we give guarantees on worst-case regret; though our assumptions are specific, we are optimistic about principled approaches that extend to more general settings and guarantee safety of decision making agents. Our computational study was constrained by the resources available—given the initial promising results of our method, we hope to scale up our experimental campaign to greater numbers of both GPUs and RLHF steps in order to see how our methods perform with larger computational budgets.

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

# A  RKHS REGRESSION

At step $t$, we have data $\{(x_1, a_1, a'_1, w_1), \ldots, (x_t, a_t, a'_t, w_t)\}$. The kernel ridge regression estimate is defined by,

$$\mu_t = \arg\min_{f \in \mathcal{H}} \sum_{i=1}^{t} (f(x_i, a_i) - w_i)^2 + \lambda \|f\|_{\mathcal{H}}^2 . \tag{13}$$

Denote by $\boldsymbol{w}_t = [w_1, \ldots, w_t]^T$ the vector of observations, $(K_t)_{i,j=1,\ldots,t} = k(x_i, a_i, x_j, a_j)$ the data kernel matrix, and $k_t(x, a) = [k(x, a, x_1, a_1), \ldots, k(x, a, x_t, a_t)]^T$ the data kernel features. We then have

$$\mu_t(x, a) = k_t(x, a)^T (K_t + \lambda \mathbf{1}_t)^{-1} \boldsymbol{w}_t . \tag{14}$$

We further have the posterior variance $\sigma_t(x, a)^2$ that determines the width of the confidence intervals,

$$\sigma_t(x, a)^2 = k(x, a, x, a) - k_t(x, a)^T (K_t + \lambda \mathbf{1}_t)^{-1} k_t(x, a) . \tag{15}$$

# B  PROOF OF THEOREM 1

In this section we will prove our main Theorem, 1. The overall strategy of the proof is to use our Lipschitz assumption on the link function (more precisely, the relative Lipschitzness of the reward $r$ and the Borda function $f_r$) in order to go to the Borda function, which we can directly model from data. Then, we use our selection criteria as well as confidence bounds taken from Chowdhury & Gopalan (2017) and convergence rates taken from Kandasamy et al. (2019) in order to complete the argument. We give these cited results as lemmas in what follows.

In order to attain a particular policy performance with probability $1 - \delta$, we must bound the error of the estimates given by our KRR process for a particular confidence level. In order to do so, we adapt the result from Chowdhury & Gopalan (2017), Theorem 2.

**Lemma 2.** *Let* $\beta_t^{(r)} = 2\|f_r\|_\kappa + \sqrt{2(\Phi_{t-1}(\mathcal{X} \times \mathcal{A}) + 1 + \log(2/\delta))}$. *Then with probability* $1 - \delta$ *we have for all time $t$ and any point* $(x, a) \in \mathcal{X} \times \mathcal{A}$,

$$|\mu_{t-1}(x, a) - f_r(x, a)| \leq \beta_t^{(r)} \sigma_{t-1}(x, a).$$

*Proof.* To prove this result, we will verify that all the conditions from Theorem 2 of Chowdhury & Gopalan (2017) hold. Recall Assumption 1 which states that $\|f_r\|_\kappa \leq B$. Next, we observe that since $a'_t \sim U(\mathcal{A})$ (independent of everything else), we have that $\mathbb{E}[w_t \mid \mathcal{F}_{t-1}] = f_r(x_t, a_t)$, where $\mathcal{F}_t = \rho\left(\{(x_s, a_s, a'_s, w_s)\}_{s=1}^t\right)$ is the filtration generated by the past observations. Additionally, since $w_t \in \{0, 1\}$ and $x_t, a_t$ are both $\mathcal{F}_{t-1}$ measurable, we see that $w_t$ can be written as

$$w_t = f_r(x_t, a_t) + \eta_t,$$

where $\eta_t$ is $\mathcal{F}_{t-1}$-conditionally subGaussian. Therefore, we have met all the necessary conditions, and we can apply Theorem 2 of Chowdhury & Gopalan (2017) which gives us the desired result. $\quad\square$

This lemma jointly bounds the modeling error over the Borda function for all time $t$ though it introduces a dependence on the RKHS norm of $f_r$. This dependence is inherited from prior work, but we empirically study the relationship between the RKHS norm of a particular reward function and that of the associated Borda function in Section C.

We also adapt a result from Lemma 8 of Kandasamy et al. (2019) in order to understand the convergence of our uncertainty function $\sigma_t$.

**Lemma 3.** *Suppose we have $n$ queries* $(q_t)_{t=1}^n$ *taken from* $\mathcal{X} \times \mathcal{A}$. *Then the posterior $\sigma_t$ satisfies*

$$\sum_{q_t} \sigma_{t-1}^2(q_t) \leq \frac{2}{\log(1 + \eta^{-2})} \Phi_n(\mathcal{X} \times \mathcal{A}).$$

Lemma 3 gives us a handle on how quickly we can expect the uncertainty function to shrink as additional datapoints are observed.

Now that we have lemmas 2 and 3 in place, we can proceed to the proof of the main result.

*Proof.* In this proof, we condition on the event in Lemma 2 holding true. Given that occurence, we can say the following for every $x \in \mathcal{X}$.

$$\max_{a \in \mathcal{A}} r(x,a) - r(x, \hat{\pi}_T(s)) \stackrel{\text{Assumption 2}}{\leq} L_1 \left( \max_{a \in \mathcal{A}} f_r(x,a) - f_r(x, \hat{\pi}_T(x)) \right) \tag{16}$$

$$\stackrel{\text{Lemma 2}}{\leq} L_1 \left( \max_{a \in \mathcal{A}} f_r(x,a) - \max_{t \in [T]} \underline{f_r^t}(x, \hat{\pi}_T(x)) \right) \tag{17}$$

$$\stackrel{\text{Def. of } \hat{\pi}_T}{=} L_1 \left( \max_{a \in \mathcal{A}} f_r(x,a) - \max_{a \in \mathcal{A}} \max_{t \in [T]} \underline{f_r^t}(x, a) \right) \tag{18}$$

$$= L_1 \min_{t \in [T]} \left( \max_{a \in \mathcal{A}} f_r(x,a) - \max_{a \in \mathcal{A}} \underline{f_r^t}(x, a) \right) \tag{19}$$

$$\stackrel{\text{Lemma 2}}{\leq} L_1 \min_{t \in [T]} \left( \max_{a \in \mathcal{A}} \overline{f_r^t}(x,a) - \max_{a \in \mathcal{A}} \underline{f_r^t}(x, a) \right) \tag{20}$$

$$\stackrel{\text{Def. of } x^t}{\leq} L_1 \min_{t \in [T]} \left( \max_{a \in \mathcal{A}} \overline{f_r^t}(x^t,a) - \max_{a \in \mathcal{A}} \underline{f_r^t}(x^t, a) \right) \tag{21}$$

$$\stackrel{\text{Def. of } a^t}{\leq} L_1 \min_{t \in [T]} \left( \overline{f_r^t}(x^t, a^t) - \underline{f_r^t}(x^t, a^t) \right) \tag{22}$$

$$\leq \frac{L_1}{T} \sum_{t=1}^{T} \left( \overline{f_r^t}(x^t, a^t) - \underline{f_r^t}(x^t, a^t) \right) \tag{23}$$

$$= \frac{L_1}{T} \sum_{t=1}^{T} 2\beta_t^{(r)} \sigma_t(x^t, a^t) \tag{24}$$

$$\stackrel{\beta_t^{(r)} \text{ is increasing}}{\leq} \frac{2L_1 \beta_T^{(r)}}{T} \sqrt{\left( \sum_{t=1}^{T} \sigma_t(x^t, a^t) \right)^2} \tag{25}$$

$$\stackrel{\text{Cauchy-Schwarz}}{\leq} \frac{2L_1 \beta_T^{(r)}}{T} \sqrt{T \sum_{t=1}^{T} \sigma_t^2(x^t, a^t)} \tag{26}$$

$$\stackrel{\text{Lemma 3}}{\leq} \frac{2L_1 \beta_T^{(r)}}{\sqrt{T}} \sqrt{C_1 \Phi_T} \tag{27}$$

$$\stackrel{\text{def of } \beta_T^{(r)}}{=} \frac{2L_1}{\sqrt{T}} (2B + \sqrt{2(\Phi_{t-1} + 1 + \log(2/\delta))}) \sqrt{C_1 \Phi_T} \tag{28}$$

$$= O \left( \frac{L_1}{\sqrt{T}} \left( B + \Phi_T \sqrt{\log \frac{1}{\delta}} \right) \right). \tag{29}$$

$\square$

## C  RKHS NORMS OF $r$ AND $f_r$

In order to understand the dependence of our estimation bound on the RKHS norm $||f_r||_\kappa$, we ran numerical experiments on sampled reward functions. For a variety of context and action dimensions, we sampled 1000 reward functions as in Section 4.3 and numerically approximated their RKHS norms. We also made a Monte-Carlo estimate of the Borda function $f_r$ for each of the reward functions sampled and numerically approximated its RKHS norm. To do this, we uniformly sample 1,000 points $x_i$ from the input space, compute the regularized kernel matrix $K$ for this set $x_i$, solve

| Context Dimension | Action Dimension | Win Rate | Win Margin |
|---|---|---|---|
| 0 | 1 | 0.16 | -6.3 |
| 1 | 1 | 0.89 | 5.1 |
| 1 | 3 | 1 | 21.4 |
| 3 | 1 | 1 | 21.5 |
| 3 | 3 | 1 | 38.7 |
| 10 | 10 | 1 | 19.6 |

Table 1: Comparison of RKHS norms of reward functions and associated Borda functions

the KRR problem $K\alpha = f(x)$ for $\alpha$. Then we compute the quadratic form $\sqrt{\alpha^T K \alpha}$ as an estimate of the RKHS norm.

In Table 1, we present the results of comparing the RKHS norms of 1000 reward functions and their associated Borda functions sampled as in Section 4.3. A 'win' was counted when the Borda function had smaller RKHS norm and a 'loss' otherwise. The win margin is the average difference in RKHS norms of the reward and Borda functions, with a positive value when the Borda function was of smaller norm. It is clear here that in general (though not always) the RKHS norm of the Borda function $f_r$ for a particular reward function $r$ is smaller than the RKHS norm of the reward function $r$ itself. This relationship seems to grow stronger as the input dimensionality of the reward function grows larger.

## D    ADDITIONAL EXPERIMENTS FOR KERNELIZED SETTING

In Figure 5, we depict the progress of the AE-Bordamethod as it continually acquires data. One can see that the estimated optimal policy (red, second row) converges to a function quite similar to the ground truth (red, first row) as more data is collected. In addition, it is clear that the selection criterion targets parts of the domain which are relevant to policy learning while avoiding obviously bad regions. We also see in the fourth row that the uncertainty over the value function decreases relatively smoothly across the context space, supporting the idea that our method controls max-regret effectively.

## E    THE JEOPARDY! PREFERENCE DATASET

We generated a set of plausible wrong answers for the Jeopardy! dataset from Huggingface (Wolf et al., 2023) by asking GPT-3.5 for a plausible wrong answer given the question, category, and answer. We found that both the category and correct answer were necessary to include to direct GPT-3.5 to generate an answer which was appropriate for the category and to prevent it from accidentally generating a correct answer. We give the prompt used for this process in Figure 6.

## F    RELATED WORK ON UNCERTAINTY ESTIMATION IN LARGE LANGUAGE MODELS

Estimating the epistemic uncertainty in large language models is still an active area of research and there are few prior works on this topic. For example, Osband et al. (2022) augment existing models with additional layers to model randomness, and subsequently the uncertainty. However performing uncertainty quantification in a parallelized fashion requires a significant memory overhead. To be more amenable to larger models, we instead use a dropout-augmented model to estimate uncertainty, as detailed in Section 5.

## G    PROMPT TEMPLATES

The prompt templates for GPT-4 as the pairwise comparison evaluation judge and GPT-3.5 as the Jeopardy! single answer correctness judge are listed in Figures 7 and 8. We maintain the standardized prompts proved to be effective by Zheng et al. (2023).

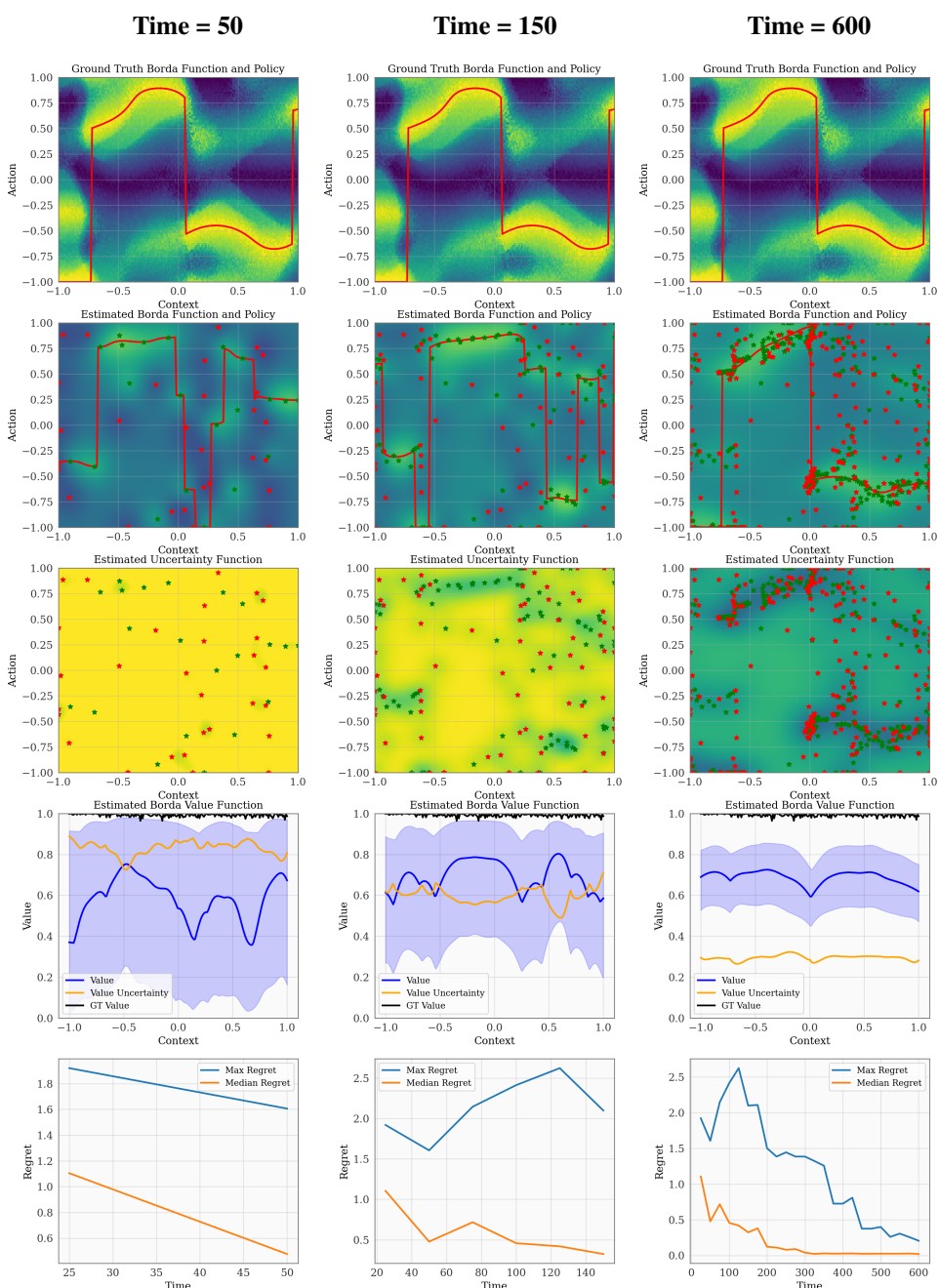

Figure 5: Progress of AE-Borda across 50, 150, and 600 datapoints. From the top downwards, the charts show the ground truth function, the mean of the posterior estimate of $f_r$, the uncertainty function, the estimate of the value function as well as the acquisition function given in Eq. (3), and the regret over time.

```
[System]
You are an assistant coming up with plausible but incorrect answers
 to Jeopardy questions (just the answer, no "what is"). Here's an
example:\n
Q: 'For the last 8 years of his life, Galileo was under house
arrest for espousing this man's theory'
Category: HISTORY
Correct Answer: Copernicus\n
Response: Brahe
[User]
Q: {question}
Category: {category}
Correct Answer: {answer}
Response:
```

Figure 6: The prompt used to collect plausible wrong answers for Jeopardy! questions.

```
[System]
Please act as an impartial judge and evaluate the quality of the
responses provided by two AI assistants to the user question
displayed below. You should choose the assistant that follows the
user's instructions and answers the user's question better. Your
evaluation should consider factors such as the helpfulness,
relevance, accuracy, depth, creativity, and level of detail of
their responses. Avoid any position biases and ensure that the
order in which the responses were presented does not influence your
 decision. Do not allow the length of the responses to influence
your evaluation. Do not favor certain names of the assistants. Be
as objective as possible. Output your final verdict by strictly
following this format: 'A' if assistant A is better, 'B' if
assistant B is better, and 'C' for a tie. Output only that
character and do not include any other characters or spaces.

[User Question]
{question}

[The Start of Assistant A's Answer]
{answer_a}
[The End of Assistant A's Answer]

[The Start of Assistant B's Answer]
{answer_b}
[The End of Assistant B's Answer]
```

Figure 7: The default prompt for pairwise comparison.

```
[System]
You are a judge on whether a contestant answer to Jeopardy is
correct given a correct answer. If you don't see the correct answer
 it is not correct. Answer 'Yes' or 'No' is sufficient. Please don'
t use any other words.

[The Start of Correct Answer]
{correct_answer}
[The End of Correct Answer]

[The Start of Contestant Answer]
{contestant_answer}
[The End of Contestant Answer]
```

Figure 8: The default prompt for evaluating single Jeopardy! answer.

## H   ADDITIONAL EXPERIMENT DETAILS

We train our initial SFT models for 1 epoch on the SHP and HH dataset and 2 epochs on the new Jeopardy! dataset. We select the initial training period based on the amount of training after which we obtained a validation loss which had plateaued. We also find it reasonable to add a dropout layer before the penultimate linear layer since we find that adding a dropout layer not to negatively affect the performance in the SFT phase. To aid in fitting the model on our GPUs, we use QLoRa (Hu et al., 2021; Dettmers et al., 2023) with 4bit quantization for model weights and optimize using the 8-bit Lion optimizer (Chen et al., 2023). For the methods with a reference model, we put the policy and the reference model on two separate GPUs. Further, we use dropout probability of $p = 0.05$, policy constraint weight $\gamma = 0.1$, an uncertainty bonus $\beta = 4$, a learning rate of $5 \times 10^{-7}$, an unlabeled batch size of 128, and a training batch size $b$ of 32. We run all experiments with 3 random seeds. Our implementation was built atop the one provided by the authors of the DPO paper (Rafailov et al., 2023).

## I   EXPERIMENT RUNTIMES

|  | Jeopardy! | SHP | HH |
|---|---|---|---|
| Further SFT | 2 \| 3 | 4 \| 4 | 3 \| 7 |
| DPO | 7.5 \| 25 | 10 \| 14 | 10 \| 15 |
| US-DPO | 8 \| 12 | 79 \| 85 | 31 \| 85 |
| AE-DPO | 9 \| 12 | 44 \| 45 | 18 \| 53 |

Table 2: Runtimes (min | max) for each experiment rounded to nearest hour. Several experiments require a significant amount of compute time to complete. Runtimes vary depending on current loads on compute clusters.

## J   ADDITIONAL EXPERIMENTS WITH LLM

Here, we plot the training curves for the Jeopardy! dataset below. For Jeopardy!, we plot the correctness of the policy over time in Figure 9. Though this is part of the goal of the agent in the Jeopardy! dataset, note that it is not the entire optimization objective, as we show in Figure 4. Here, it is clear that no policy is able to improve at predicting correct answers on the test set. This is unsurprising as trivia is a difficult generalization problem.

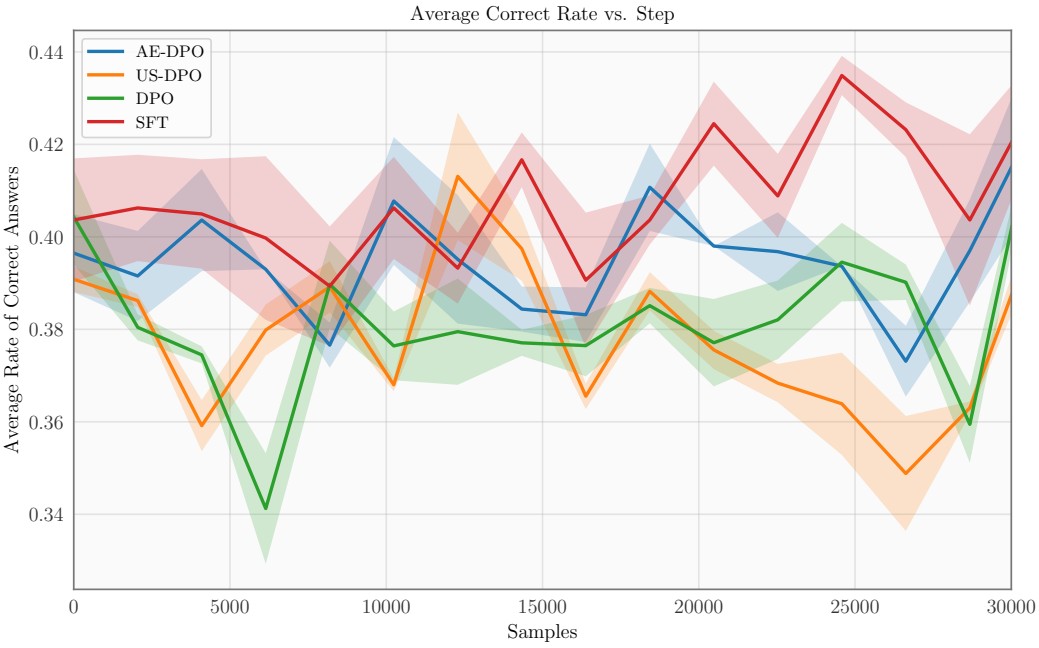

Figure 9: Rate of correct answers for Jeopardy! over time.

## J.1    Evaluating dropout-based LLM uncertainty estimation

We believe that in general the estimation of uncertainty for LLMs is an important topic of research and progress there will facilitate a more efficient and informed use of this technology. As we discussed in appendix F and section 5, we use a dropout-based uncertainty estimation technique to inform the active exploration in this work. Over the course of this study, we considered ensembles and epistemic networks (Osband et al., 2022) as alternative methods for estimating the uncertainty of LLMs. However, each of these methods comes with some additional GPU memory requirement. For epistemic networks, the additional network parameters take GPU memory, while for ensembles, the memory is required to store multiple copies of a network or at least mutiple LoRAs. In our initial studies we found epistemic networks and dropout to perform comparably well and therefore chose dropout due to its smaller memory consumption and good performance. In this section, we explore whether the uncertainties predicted by our estimates differ when the model predicts the correct, incorrect, or null answer and whether these predictions differ in the cases when the model decides to predict null. To do this, we evaluated the log probabilities predicted by $\pi_{\text{SFT}}$ on a test set of 20,560 Jeopardy! clues for the correct, incorrect, and null answer. We computed the sample variances over the log probabilities $\sigma^2(a \mid x) = \sum_{t_i \in a} \sigma^2(t_i \mid x, t_1, \ldots, t_{i-1})$ and plotted their densities in fig. 10.

We see that the model predicts the highest variances for the log probabilities of incorrect answers. We also see that the the model seems to predict especially low variances for the null token when it decides to output it. The correct answer seems to have a lower variance when the model is willing to predict an answer. We see that the log probabilities of incorrect answers always have a high variance, indicating high uncertainty. We also see that the null token has a low variance when the model has a non-null output indicating certainty that it should not abstain. The variance further drops when it outputs null, indicating certainty about not knowing an answer. The correct answer has a lower variance than the incorrect answer when the model does not abstain. The relative variances of these two curves support that the model provides meaningful indications of uncertainty. Additionally, in the case where the model abstains, even the correct answer has a high variance, indicating a high uncertainty. We believe that these results support that the uncertainty function is at least correlated with the model's knowledge about the input. This offers support to the hypothesis that our estimates of the variance are somewhat meaningful. However, we believe that this is an important research

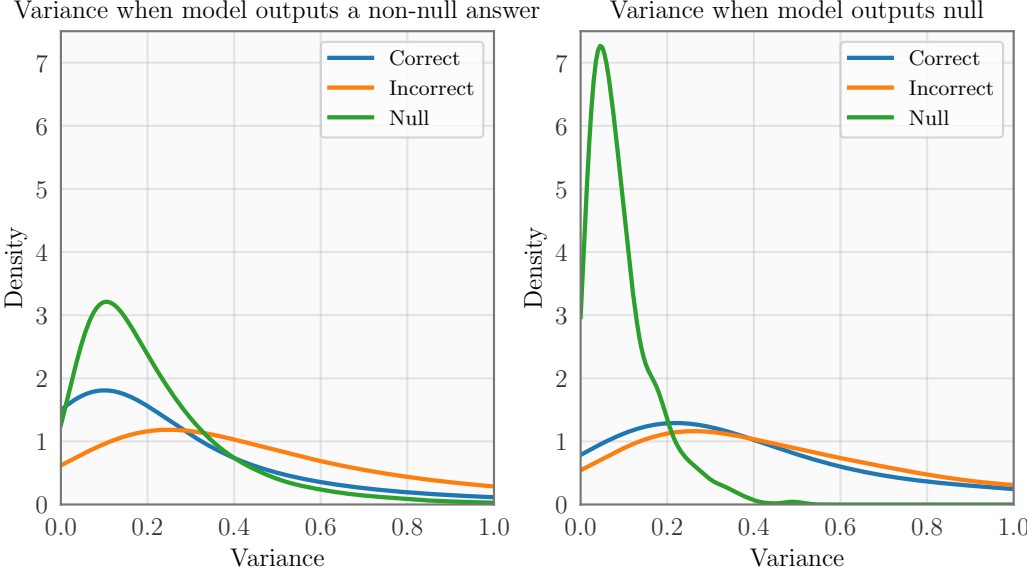

Figure 10: Density of $\sigma(a \mid x)$ conditioned on correct, incorrect, and null values for $a$. The left hand plot depicts the variance distributions conditional on the model outputing a non-null completion, while the right hand is conditional on a null completion.

topic and warrants substantial further study under a variety of lenses. We hope that this work will encourage further research in this area.

