# OpenReview forum: "Sample Efficient Reinforcement Learning from Human Feedback via Active Exploration"
_ICLR.cc/2024/Conference — Submitted to ICLR 2024_

### Official Review · Reviewer_NrDs · 2023-10-20

**Soundness:** 3 good
**Presentation:** 3 good
**Contribution:** 2 fair
**Rating:** 5
**Confidence:** 3

**Summary:**

This paper studies active learning in comparison-based contextual bandit. The authors propose a new approach for choosing contexts and actions effectively in order to learn a policy that has low suboptimality in the worst context. A theoretical guarantee on the suboptimality of the basic version of the algorithm (AE-Borda) is provided under RKHS. Extensive empirical results are provided for a DPO version of the algorithm (AE-DPO), including some experiments on Llama-7B.

**Strengths:**

The paper contains a theoretical result as well as extensive empirical results. The theoretical section offers a new perspective about RLHF with the Borda function. The plots are nicely made.

**Weaknesses:**

It would be nice if there is more discussion about why the proposed algorithm has better performance than existing algorithms (e.g., provide some intuition).

In Section 5.1, AE-DPO is compared with US-DPO, DPO, SFT. Among these baselines, DPO and SFT are really offline methods with uniform samples and not active learning methods, so US-DPO is the only active learning method that can compete with AE-DPO. Comparison with only one other active learning method seems slightly insufficient. Is there any other active learning algorithms in the existing literature that can be compared to?

**Questions:**

- Algorithm 1 relies on $\Phi_t$. How do we compute this quantity that depends on $r_A$, which I don't think is accessible in this setting?

- Many existing comparison-based algorithms learn the reward function first and then use it to construct the policy. In contrast, Algorithm 1 chooses to learn the Borda function. What is the advantage of learning the Borda function over learning the reward function?

- Both the abstract and Section 3 say the problem is offline contextual bandit, but I don't think the learner can freely query contexts and actions in the offline setting. In fact, wouldn't the ability of freely querying contexts and actions make this problem too simple? In this setting, the problem is just learning a function with a dataset chosen by the learner; the only difficulty is the data is pairwise comparisons.

- In Section 5.1, the authors observed that on Jeopardy! dataset, the policy trained with AE-DPO is able to abstain when it is not expected to know the correct answer, in contrast to policies trained with other baseline methods. Should this reduction in hallucination be attributed to your algorithm or just the objective you are using (Equation (2)), which is also supposed to make the learned policy behave more prudently and abstain when it is likely to answer incorrectly?

I'm open to raising my score after reading the clarification from the authors and their discussion with other reviewers during rebuttal.

---

> ### Author Response · Authors · 2023-11-16
>
> Thank you for your thoughtful review and especially for your questions. We hope to use the feedback to improve the paper. Here, we’ll first answer your questions then address the weaknesses you pointed out.
>
> ### **Questions**
> We agree that $\Phi_t$ is an important quantity in our algorithm as in many similar kernelized methods [1][2] and also that computing it is infeasible in practical situations. We only wish to clarify that $\Phi_t$ is important for the analysis and not the implementation of our algorithm; we use known worst-case bounds on this quantity to give concrete performance bounds in Section 4.2 without having to know more about the specific value of $\Phi_t$.
>
> When we initially attempted to address active RLHF, we considered approaches that involved directly estimating the reward function. However, it was difficult to design a principled strategy for choosing a context and pair of actions when we could only evaluate a single context-action pair at a time. By “marginalizing” the second action through the uniform sampling strategy we were able to estimate the Borda function, which implicitly involves the second action and thereby allows us to determine a principled strategy for active exploration in our setting.
>
> In writing this paper, we have followed [3] in viewing the term “offline” in bandits or BO as a setting where the agent can adaptively choose a context and a pair of actions and observe a winner (this setting could be useful in contexts where e.g. there is a paid labeler giving preferences over completions given a prompt for language model alignment). However, we understand that the “offline” name is confusing and have edited it throughout the paper to “active” as you suggest. In fact, we believe the key contribution of our paper both in the theoretical and practical setting is in the active learning methods rather than in estimation or policy learning.
>
> Our method was designed to be effective in the worst case over contexts (equation 2) and chooses contexts in order to optimize that objective. We believe that the performance of the AE-DPO policy in abstaining on the Jeopardy dataset is at least partially due to some success in optimizing the objective and therefore would attribute that success to the method at least in part.
>
>
> ### **Weaknesses**
> We agree that the DPO and SFT baselines are not active learning methods but are suitable for static datasets. To our knowledge there have not been works addressing the question of how to actively select data for RLHF; we therefore are limited for comparison to this relatively simple set of baselines (and please feel free to comment with a citation if you think that we have missed a relevant prior work). We do hope that others propose even better active RLHF methods in the future as we think this problem setting is important.
>
> Thank you once more for your constructive comments on our work. In response, we have revised our manuscript, and hope we have addressed your concerns adequately. If you believe our updates and responses are satisfactory, an adjustment in your score would be highly valued. Please feel free to let us know any additional comments or concerns.
>
> [1] https://arxiv.org/abs/2011.04622
>
> [2] https://arxiv.org/abs/2212.09510

---

### Official Review · Reviewer_DPyN · 2023-10-28

**Soundness:** 3 good
**Presentation:** 3 good
**Contribution:** 3 good
**Rating:** 6
**Confidence:** 2

**Summary:**

The paper proposes an adaptive exploration method for RLHF. The method is based on a UCB style approach which selects the context that maximizes the gap between optimistic and pessimistic estimations of the Borda function. The paper then performs LLM experiments to justify the advantage of the adaptive exploration method.

**Strengths:**

The paper has good writing and is easy to follow. The theoretical formulation is clean and the result is solid.

**Weaknesses:**

1. One weakness is the theoretical part is less related with the LLM experiments. The theory is associated with Borda function and RKHS, but in the LLM part both concepts are removed. And while AE-Borda is a value-based algorithm, the LLM part switches to a policy-based algorithm instead. The only shared idea is both algorithms select the context which maximizes some optimistic gap. But the link between theory and experiment is still weak.

2. In Figure 4, the result of AE-DPO tends to have a higher variance compared with other baselines. This fact could make the paper's claim less convincing as it's possible that the plot happens to choose the good seeds, given that the result is so noisy.

**Questions:**

For the Jeopardy dataset, one may find that the null rate for incorrect answer starts to decrease when the number of samples further goes up, which means that large sample size is not always helpful. This is very different from the conclusion of the theory part. Any comments on this fact?

---

> ### Author Response · Authors · 2023-11-16
>
> Thank you for your detailed review and the valuable questions you've posed. Below we address each of your questions and comments.
>
> Per your Jeopardy question and your comment on the variance of AE-DPO methods: we had a similar feeling at submission time, and were generally curious about the performance of AE-DPO if it were run longer. So, we re-ran all LLM experiments with a larger compute budget (up to 100k samples) and have included the additional results in our updated draft. For the specific jeopardy findings we believe that the general trend is supported by the additional evidence though the learning curve is not strictly monotonic. We welcome any ideas or research that can help us better understand these learning dynamics. We also believe that though there is still high variance across our seeds (we would have liked to run many more but are limited due to compute budget), our results hold up to further scrutiny.
>
> To your point on the connection between the kernelized and LLM sections; we agree that there are differences between the methods. To be explicit, the connection between the first and second method is that the first method gives theoretical support for choosing the context for which the “value” of the optimal policy is most uncertain and then acting optimistically, and the second method attempts to implement the same intuitive rule when subjected to the constraints of the DPO training setup. One choice we could have made to make the value function more explicit was to use the RLHF pipeline with a separate reward model as in [1]. We decided that it was better to use our policy model’s uncertainty estimates for our data selection rule than the estimates of a surrogate preference model and follow DPO in estimating state-action values from the log probabilities.
>
> We deeply appreciate the time and effort you invested in reviewing our submission. Our updated draft and the accompanying responses aim to address your observations and recommendations effectively. If we’ve met your expectations, we ask that you please consider improving your score. We welcome any further thoughts or questions.
>
> [1] https://arxiv.org/abs/2203.02155

---

### Official Review · Reviewer_aHAt · 2023-10-30

**Soundness:** 2 fair
**Presentation:** 2 fair
**Contribution:** 2 fair
**Rating:** 3
**Confidence:** 5

**Summary:**

This paper studied an interesting problem: how to actively collect data during reward model learning. The authors cast the problem into a contextual dueling bandit problem and proposed an algorithm with regret bound. Some experiments are conducted for the full LM case.

Overall, I feel this work studied a timely topic but was not well executed. The current context may not be sufficient to be accepted at a top-tier ML conference. I evaluated this work from two aspects: theoretical contribution and empirical contribution.

For theoretical contribution, I feel it is rather limited. First, contextual dueling bandit has been studied for a while in the bandit community. If one only cares about worse-case simple regret bound, any algorithm that enjoys a cumulative regret guarantee can be turned into a simple regret minimization algorithm, for example, "Optimal Algorithms for Stochastic Contextual Preference Bandits". Second, Assumption 2 is quite strong. It is very hard to satisfy for large action space which is the case for LM. The theory largely benefits from this assumption. I do not believe any interesting LM application can satisfy this.

Second, the algorithm proposed in Section 5 is very different from the one in previous sections with guarantee and the algorithm is very heuristic. The authors seem to use ensemble dropout to estimate the standard deviation. This is very doubtful if dropout can estimate the variance well for an autoregressive transformer. As far as I know, there has been no study on that before. More importantly,  none of the win-rate is statistically significant, especially. for the Anthropic dataset. It is hard for me to trust any conclusion from such a noisy result.

Minor: 1. Why the win-rate is far below 0.5 in Figures 9 and 10? I suppose the baseline is uniform sampling.
2. The term 'offline' contextual bandits is very misleading. I think you are doing online learning: actively collect human feedback. Offline problem usually refers to the case the dataset is given.
3. In Algorithm 1, the second action is drawn uniformly random. This is weird and why it could work? Will this benefit from Assumption 2 as well?
4. DPO also has experiments on the Anthropic dataset. You should at least report or discuss the win-rate matched in their setting to make sure if the implementation is correct.
5. How do you generate multiple completions?

**Strengths:**

See summary.

**Weaknesses:**

See summary.

**Questions:**

See summary.

---

> ### Author Response · Authors · 2023-11-16
>
> Thank you for your candid feedback. We've taken your points to heart and aim to clarify these areas. Below, we respond to your questions and outline revisions that we hope will address your concerns effectively.
>
> ### **Questions on Theory**
> We agree that our theoretical techniques have been developed for other problem settings. However, we want to emphasize that the contribution of our paper is not in novel analysis techniques but rather in a new algorithm, choice of problem setting, and LLM extension that addresses a topical problem and progresses the state of the art there.
>
> We also want to emphasize that existing works do not directly give sample complexity guarantees for our setting. This is due to the fact that we work under a much stronger notion of sub-optimality (sub-optimality of the policy for the worst case context) than existing works which look at the sub-optimality when the contexts are drawn from some distribution. While it is true that cumulative regret minimization algorithms can be converted into simple regret minimization algorithms, the analog in our setting would require algorithms that have small cumulative worst case regret (as opposed to the algorithm you reference in [1] which has small cumulative expected regret). We do not believe that such algorithms exist and would welcome citations that show otherwise.
>
> Assumption 2 we believe is a quite weak assumption. We added some clarifying text on it to the paper spelling out that the most commonly used link functions, the logistic function and Gaussian CDF, satisfy our assumption. In fact, for the logistic link function used in most LLM applications, we know concretely that setting $L_1= 2$ is sufficient for our assumption to hold.
>
> ### **Questions on Practical Method**
>
> We agree that uncertainty estimation is a key component of our method. In order to address some of your concerns about this we have included an additional small study of the effectiveness of our dropout-based method in section J.1 that helps to justify our choice. As we developed our method, we considered ensemble-based methods as well as epistemic networks as in [2]. However, we found that each of these seemed to substantially increase the memory footprint of model training and in our preliminary studies it seemed that dropout and epistemic networks had similar performance, which was better than that of the ensemble. Therefore, we went with the simplest choice of dropout for our uncertainty. We added a note mentioning this to Section J.1 as well. We welcome further work studying uncertainty quantification in LLMs as we agree it is understudied and would benefit our efforts.
>
> To address your comments about statistical significance of our results, we re-ran all our LLM experiments for longer (100k datapoints rather than 30k) and added the updated results to the experiments section. As you can see from these charts (in Figure 4), the sample efficiency of the AE-LSVI is clearer over the longer duration of these updated runs.
>
> ### **Minor Points**
>
> Minor 1. To compute the win-rate, we compare generated output against  the preferred completions taken from the Anthropic / SHP datasets, which are the best examples we can find of human preferences on those tasks. This should not be expected to be greater than 0.5 win-rate, as a bad performance here could go as low as 0.0 win rate – it is likely that a random sampling baseline would obtain a win-rate far below 0.5, since randomly sampling tokens would result in incoherent generations.
>
> Minor 2. We agree that “offline” was a bad choice of terminology and have updated it to “active” throughout the paper.
>
> Minor 3. The choice of a uniform second action in Algorithm 1 follows [3] and allows us to directly estimate the Borda function over a context and a single action from the preference labels collected. This is essential to the functioning of our method. The theoretical justification for this does indeed depend on Assumption 2, but as we mentioned above we do not see it as overly restrictive. Our rates of convergence seem to match those attainable in the non-dueling setting [4] so it doesn’t seem to affect the theoretical performance.
>
> Minor 4. As seen in the DPO paper [5] it is possible to exceed a 50% win rate in the best case, but not by much. Our setup is more challenging than that of the DPO in that we do not first do supervised fine-tuning on the entire training set but instead use a 30% split. This is due to the fact that we wanted to test our data selection policy on “fresh” data and provide the remaining 70% as the batch from which we could select. As we started from an implementation provided by the authors of DPO we believe that our DPO implementation is correct. Our results seem consistent with those in Figure 3 in the DPO paper and to verify this, we began by reproducing these results.

---

> > ### Author Response · Authors · 2023-11-16
> >
> > Minor 5. For training we use multiple completions taken from the datasets provided, while at test time (ori in a truly online trial) we generate from sampling autoregressively from our language model policy.
> >
> > Thank you for your valuable input regarding our paper. We have aimed to incorporate each of your suggestions in our revised draft. Should you find that these modifications meet your expectations, we would appreciate a re-evaluation of your initial score. We are happy to answer any further questions or clarifications.
> >
> > [1] https://arxiv.org/abs/2111.12306
> >
> > [2] https://arxiv.org/abs/2107.08924
> >
> > [3] https://arxiv.org/abs/1911.00980
> >
> > [4] https://arxiv.org/abs/2212.09510
> >
> > [5] https://arxiv.org/abs/2305.18290

---

### Official Review · Reviewer_vRQm · 2023-10-30

**Soundness:** 3 good
**Presentation:** 3 good
**Contribution:** 2 fair
**Rating:** 5
**Confidence:** 3

**Summary:**

This paper takes advantage of the fact that one can often choose contexts at which to obtain human feedback in order to most efficiently identify a good policy, and formalizes this as an offline contextual dueling bandit problem. This paper proposes an upper-confidence-bound style algorithm and proves a polynomial worst-case regret bound. Then, the authors provide empirical confirmation in a synthetic setting that their approach outperforms existing methods, and further extend the setting and methodology for practical use in RLHF training of large language models (LLMs).

**Strengths:**

1.	The studied problem, i.e., offline contextual dueling bandit with human feedback, is well-motivated and models the RLHF training of large language models.
2.	The paper provides extensive experimental results in both synthetic and practical LLM settings.

**Weaknesses:**

1.	The techniques used in the proposed algorithms, e.g., the estimation of the Borda score by uniform sampling, active learning and confidence intervals, are well-known. The authors should elaborate more on the technical novelty.
2.	The procedures of Algorithms 1 and 2 are not clear. It would be better to specify the definitions of $\mu_t(x,a)$ and $\sigma_t(s,a)$ in the main text. The notation $\sigma_t(x,a)$ is overlapped with the notation of link function $\sigma$.
3.	Can the authors compare their algorithms with the MLE method for learning the reward model, and discuss the advantages of their algorithms?
4.	It seems that Algorithms 1 and 2 need to compute the argmax operation over the context space $\mathcal{X}$ and the action space $\mathcal{A}$. Can these algorithms be extended to the large context and action space setting? In LLMs, the spaces of contexts and actions are often large.

---

**---After Rebuttal---**

Thank the authors for their rebuttal. I read the authors' rebuttal and other reviewers' comments.

In my opinion, while the authors consider a stronger (variant) notion of suboptimality, the theoretical part (Section 4) of this paper is not novel, since the ideas of estimating Borda score and selecting the option with the maximum uncertainty is well-known in the dueling bandit and active learning literatures. I think the more interesting contributions of this paper are the well-motivated problem formulation which is applicable to LLMs, and the experiments on LLMs with the proposed algorithm. However, to some degree I agree the comments of Reviewers 7uH1 and aHAt, i.e., the algorithm in Section 5 is heuristic and a little disconnected with the theoretical results in Section 4. The algorithm design and empirical results for LLMs (Section 5) seem to lack the theoretical supports.

I tend to keep my score 5, and will listen to the opinions of other reviewers and AC during the discussion period.

**Questions:**

Please see the weaknesses above.

---

> ### Author Response · Authors · 2023-11-16
>
> We appreciate your insightful review and the questions you've raised. Your feedback is crucial for us to improve our paper. In this response, we will begin by addressing your questions, followed by a discussion on the areas of improvement you've highlighted.
>
> ### **Questions on Novelty of Contributions**
> We agree that our theoretical techniques have been developed for other problem settings. However, we want to emphasize that the contribution of our paper is not in novel analysis techniques but rather in a new algorithm, choice of problem setting, and LLM extension that addresses a topical problem and progresses the state of the art there.
>
> We also want to emphasize that existing works do not directly give sample complexity guarantees for our setting. This is due to the fact that we work under a much stronger notion of sub-optimality (sub-optimality of the policy for the worst case context) than existing works which look at the sub-optimality when the contexts are drawn from some distribution. We do not believe that such algorithms exist and would welcome citations that show otherwise.
>
> ### **Questions about Design Choices**
> We have edited the $\sigma$ for the link function to $\rho$ everywhere in the paper for notational clarity; thank you for the suggestion. We’ve also added definitions for $\mu$ and $\sigma$ in Section 5.
>
> It is possible, as you suggest, that if we fix the link function we could estimate the MLE for the reward function. We could then analyze confidence intervals over that estimate and discuss convergence. However, we could not find a natural way to choose contexts based on the estimates, while our method of estimating the contextual Borda function comes with a fairly natural algorithm and analysis afterward.
>
> In Section 5 we could have also learned a reward model with a ranking loss and then optimized the policy against it as is done in [1]. However, due to the performance, simplicity, and robustness shown by DPO and also because we wanted to use the uncertainty estimates of the policy to drive data selection, we went with a direct policy learning strategy instead.
>
> In theory, it would certainly be difficult to compute the exact argmax over a large context and action space.  However, in Algorithm 2 we show the performance of our method which approximately computes this argmax and find that it results in improved performance for the large spaces of sequences native for LLMs. We also note that this argmax problem is common to many RL or bandit algorithms with large state and action spaces and these methods often use similar approximations as what we show in our paper. [2]
>
> We are grateful for your insightful feedback and suggestions on our submission. Our revised draft aims to comprehensively address these points. If you find our revisions and responses satisfactory, we kindly request you consider improving your score. We remain open to any additional feedback you may have.
>
> [1] https://arxiv.org/abs/2203.02155
>
> [2] https://arxiv.org/abs/1512.07679

---

### Official Review · Reviewer_7uH1 · 2023-11-02

**Soundness:** 2 fair
**Presentation:** 3 good
**Contribution:** 2 fair
**Rating:** 5
**Confidence:** 2

**Summary:**

The paper studies RLHF in a setting where one is allowed to choose contexts in which feedback can be obtained. The authors develop an upper-confidence-bound style algorithm in this setting that enjoys a regret guarantee. They also show favorable empirical results on synthetic and real-world datasets in aligning language models.

**Strengths:**

- The paper makes empirical improvements toward an important topic of increasing the efficiency of RLHF, which is relevant, particularly for LLMs.
- Empirical evaluations show improvements in efficiency compared to prior work.

**Weaknesses:**

- Novelty in problem selection (i.e., the setting where the contexts can be chosen instead) and the algorithm design are limited.
- Theoretical contribution of the paper is limited to strong assumptions and the analysis techniques exist in prior works.
- The main algorithm requires uncertainty quantification for the policy, which is difficult for LLM policies. A method based on dropout is used for such uncertainty quantification; however, why this method is used over alternatives is not discussed.

**Questions:**

See weaknesses above.

---

> ### Author Response · Authors · 2023-11-16
>
> Thank you for your review and for the thoughtful questions. Your feedback is helpful for us to improve our paper. We respond back to each of your questions below and then focus on improving the aspects you've identified as needing attention.
>
> ### **Questions on Problem Setting**
> As we mentioned in the introduction, in this work we take advantage of the fact that we can control which prompts and completions we provide to human labelers in order to make efficient use of their efforts. We know that OpenAI spends 8 figures on preference data and that there are multiple startups that provide RLHF as a service. If we could make these processes more efficient this would result in substantial savings throughout this ecosystem.  Though we agree there are a substantial number of works that tackle active learning in various related settings, we were unable to find any that addressed this setting in particular. Therefore, we believe it was well worth working out the theory and practical implementations and that it will be valuable to the community to present this information.
>
> ### **Questions on Theoretical Contribution**
> Assumption 1 we believe is crucial in order to achieve the generality obtained by our algorithm. In relation to past works [2], our assumption differs only in the fact that we assume that the Borda function is also smooth in an underlying RKHS – the assumption that the reward function is smooth in an underlying RKHS is standard in the literature. We make this assumption in order to allow our algorithm to work with a user-specified link function – existing works only deal with the case when the link function is a sigmoid. Additionally, we show in appendix section C that this assumption seems to be empirically true for our synthetic examples.
>
> Assumption 2 we believe is a relatively weak assumption. We added some clarifying text on it to the paper spelling out that the most commonly used link functions, the logistic function and Gaussian CDF, satisfy it. In fact, for the logistic function used in most LLM applications, we know concretely that setting $L_1= 2$ is sufficient for our assumption to hold.
>
> ### **Questions on Uncertainty Estimation**
> We agree that uncertainty estimation is a key component of our method. In order to address some of your concerns about this we have included an additional small study of the effectiveness of our dropout-based method in section J.1 that helps to justify our choice. As we developed our method, we considered ensemble-based methods as well as epistemic networks as in [1]. However, we found that each of these seemed to substantially increase the memory footprint of model training and in our preliminary studies it seemed that dropout and epistemic networks had similar performance, which was better than that of the ensemble. Therefore, we went with the simplest choice of dropout for our uncertainty. We added a note mentioning this to Section J.1 as well.
>
> Thank you again for the feedback and suggestions in response to our submission. We hope that our updated draft and responses address them satisfactorily. We ask that you please consider increasing your score if you agree, and please let us know if you have any further questions.
>
> [1] https://arxiv.org/abs/2107.08924
>
> [2] https://arxiv.org/abs/2011.04622

---

### Meta-Review · Area_Chair_EEJM · 2023-12-05

**Metareview:**

This paper investigates using active learning for data acquisition in reinforcement learning from human feedback. It formalizes the problem as an offline contextual dueling bandit problem and provides theoretical and experimental results.

I recommend rejecting this paper for the following three main reasons:
- There seems to be limited novelty of the theoretical results.
- The assumptions made in the theoretical section appear to be too restrictive.
- The used uncertainty quantification method seems to be heuristic and overly simplistic.

**Justification For Why Not Higher Score:**

I recommend rejecting this paper for the following three main reasons:
- There seems to be limited novelty of the theoretical results.
- The assumptions made in the theoretical section appear to be too restrictive.
- The used uncertainty quantification method seems to be heuristic and overly simplistic.

**Justification For Why Not Lower Score:**

N/A

---

### Decision · Program_Chairs · 2024-01-16

Reject